

1   Quantifying thermohaline circulations: seawater isotopic compositions

2   and salinity as proxies of the ratio between advection time and

3   evaporation time

6              Hadar Berman, Nathan Paldor* and Boaz Lazar

8          The Fredy and Nadine Herrmann Institute of Earth Sciences

9              The Hebrew University of Jerusalem


*Corresponding author, Email address: nathan.paldor@mail.huji.ac.il



Abstract
Uncertainties in quantitative estimates of the thermohaline circulation in any particular basin
are large, partly due to large uncertainties in quantifying excess evaporation over precipitation
and surface velocities. A single nondimensional parameter, $\gamma \equiv \dfrac{q}{h}\dfrac{x}{u}$ is proposed to
characterize the "strength" of the thermohaline circulation by combining the physical
parameters of surface velocity ($u$), evaporation rate ($q$), mixed layer depth ($h$) and trajectory
length ($x$). Values of $\gamma$ can be estimated directly from cross-sections of salinity or seawater
isotopic composition ($\delta^{18}O$ and $\delta D$). Estimates of $\gamma$ in the Red Sea and the South-West Indian
Ocean are 0.1 and 0.02, respectively, which implies that the thermohaline contribution to the
circulation in the former is higher than in the latter. Once the value of $\gamma$ has been determined
in a particular basin, either $q$ or $u$ can be estimated from known values of the remaining
parameters. In the studied basins such estimates are consistent with previous studies.



## 1. Introduction


The thermohaline circulation is driven by net evaporation-precipitation from the surface layer,
which is compensated for by horizontal advection of (typically) small speeds. This circulation
has been used to describe the circulation in various parts of the world ocean including semi-
enclosed basins (such as the Red Sea). Due to the small horizontal velocities and excess of
evaporation (i.e. evaporation minus precipitation) it is difficult to directly quantify the
thermohaline circulation in terms of the mean water speed and/or the water column transport.
The aim of this paper is to propose a single non-dimensional parameter that quantifies the
advection vs. evaporation (referred later as the "strength") of the thermohaline circulation.

While mean velocities are small and hard to measure (since they do not differ from

their instantaneous variability), net evaporation over land has been measured for decades by
dish-pan evaporation experiments, in which the level of water in a pan (or shallow lagoon)
located on a coast is monitored over several weeks/months. Over sea however, these
experiments are not accurate since they cannot generate the microclimate of a large semi-
enclosed basin or ocean. For example, values of 3.5 m·y$^{-1}$ (Sofianos et. al, 2002; Vercelli,
1925) and 3.1 m·y$^{-1}$ (Cohen et al., 1977) were found for the Red Sea, both of which turned
out to be too high (compared with detailed heat balance calculations, Ben-Sasson et al., 2009).

Net evaporation in the ocean can be estimated only crudely from routine salinity

measurements. The reason for this uncertainty is the difficulty in separating the mean salinity
difference across the surface layer from temporal (e.g. seasonal) and spatial (i.e. the depth
above which the incoming waters flow) differences. This is why salinity based estimates of
net evaporation in a particular basin may vary by up to 100% even though the different ways
in which salinity has been measured in the last several decades have not yielded drastically
different values. For example, in the Red Sea net evaporation rates range between 1.7 m·y$^{-1}$
(Tragou, 1999) and 2.7 m·y$^{-1}$ (Sofianos et. al, 2002; Bogdanova, 1974) even though both



studies employed the same salt-conservation methodology i.e. assessing salinity differences
between incoming and outgoing waters.

Attempts to estimate the net evaporation from detailed calculations of the various

components of the heat balance in semi-enclosed basins (i.e. incoming shortwave radiation,
net outgoing longwave radiation, latent heat release and advection of heat to/from the basin)
encountered similar discrepancies. In the northern Red Sea, the detailed heat flux calculation
by Assaf and Kessler (1976) have yielded a net evaporation rate of 3.6 m·y$^{-1}$ while Ben-
Sasson et al. (2009) and Berman et al. (2003) estimated the annual net evaporation rate to be
only 1.8 m·y$^{-1}$ (see also, Sofianos, 2002).

An approach of using fractionation of stable isotopes of oxygen and hydrogen has

been previously used to trace net evaporation rates in lakes (e.g., Gibson et al., 1993; Gat et
al., 1994). The underlying mechanism is using the isotopic fractionation of oxygen and
hydrogen atoms of the $H_2O$ molecule in surface seawater during its evaporation. For
thermohaline circulation, where seawater flow is driven by net evaporation it is possible to
use the Rayleigh distillation equation (e.g. Craig et al., 1956; Appelo and Postma, 2005),
given in section 2.2 below, to describe the isotopic composition of surface seawater while
flowing and continuously loosing water by evaporation. The equation provides a simple
relation between the fraction of remaining water molecules during evaporation and the
isotopic changes in the ratio between the heavy (and rare) isotopes of $^{18}O$ and $^2H$ (deuterium,
denoted by D) and the light (and abundant) isotopes $^{16}O$ and $^1H$.

Here we derive a non-dimensional parameter which can estimate the "strength" of the

thermohaline circulation. We use three independent methods to validate this parameter:
salinity based measurements and isotopic measurements of $\delta^{18}O$ and $\delta D$ in surface waters.
We are interested in areas where excess of evaporation over precipitation dominates, in long
term time scales of multi year average, neglecting seasonal variability, and thus we looked for



areas of small horizontal velocities where the data is available to us. We found two areas
suitable for the case study, one in a semi-enclosed basin (The Red Sea) and the second in the
southwest Indian Ocean. The agreement of the independent methods will be an indirect
confirmation of the validity of the estimates of thermohaline circulation strength. The method
employs data collected from the Red Sea and the Gulf of Elat (Aqaba), and for the southwest
Indian Ocean, but can be used for any thermohaline driven water mass where the flow in the
surface layer is weak and driven by net evaporation.

## 2. Theory: Explicit expressions for the net evaporation
In this section we derive a single-parameter that quantifies the strength of the thermohaline
circulation from changes of seawater salinity (subsection 2.1) and isotopic composition of
$\delta^{18}O$ and/or $\delta D$ (subsection 2.2).

### 2.1. Conservation of salt
The simple scenario we envision, involves seawater inflowing for year(s) subject to excess
evaporation over precipitation at a rate of $Q$ m$^3 \cdot$y$^{-1}$ (e.g. when water flows from an ocean into
a semi-enclosed adjacent basin). Typical rates of evaporation worldwide are of order 1 meter
per year ($\cong 3.17 \cdot 10^{-8}$ m$\cdot$s$^{-1}$). A simple mass balance of water and salt in the basin (assuming
that density changes due to changes in salinity are second-order and can be neglected and that
salinity is conserved in the mixed layer) leads to the following relation between salinity
changes as a function of distance (similar to the classical Knudsen relation, formulated nearly
100 years ago (see Defant, 1961)):
$$S(x) \cdot \left(1 - \frac{Q}{V_0} \cdot t\right) = S(0) \tag{1}$$




where $S$ is the salinity of the mixed layer [in dimensionless units], $x$ [km] is the distance from
the starting point of the water trajectory (i.e., $x$=0 denotes Bab el Mandeb in the case of the
Red Sea), $Q$ is the excess evaporation over precipitation [m$^3$·d$^{-1}$], $V_0$ is the volume of the
mixed layer at the origin [m$^3$], $t = \dfrac{x}{u}$ is the time [d] that takes an arbitrary water volume to
pass from $x$=0 to $x$ [km] as it moves with the average speed $u$ [m·d$^{-1}$].

Substituting $Q = q \cdot A$   where $q$ is the net evaporation flux [cm·d$^{-1}$] and $A$ is the cross-

section area [cm$^2$] and letting $h = \dfrac{V(0)}{A}$ be the depth of the mixed layer transforms Eq. (1) to:
$$\frac{S(x)}{S(0)} = \frac{1}{\left(1 - \dfrac{q}{h} \cdot \dfrac{x}{u}\right)} = \frac{1}{(1-\gamma)} \tag{2}$$

where the non-dimensional parameter in Eq. (2), $\gamma \equiv \dfrac{q}{h}\dfrac{x}{u}$, which arises naturally, quantifies
the changes in salinity of the inflowing water due to net evaporation as a function of the
distance from the origin.

This parameter that combines several physical parameters associated with the

thermohaline circulation can also be written as $\gamma \equiv \dfrac{t_{adv}}{t_{evp}}$ where $t_{adv}$=$x$/$u$, is the advection time
(i.e. time it takes a water parcel with velocity $u$ to propagate the distance $x$) and $t_{evp}$=$q$/$h$, is the
evaporation time (the time it takes to completely evaporate the mixed layer).

An examination of the characteristic magnitudes of the dimensional parameters in the

Red Sea can provide an estimate for the non-dimensional model parameter $\gamma$. For $x$ = 2000 km
($\cong$2·10$^6$ m) (the length of the Red Sea) and $u$ = 1 cm·s$^{-1}$ ($\cong$10$^{-2}$ m·s$^{-1}$) (annually averaged
surface northward velocity along the sea; see figure 9 of Sofianos and Johns, 2003), assuming
a mixed layer depth (annually averaged) of $h$ = 100 m (see figure 13 of Sofianos and Johns,
2003) and a net evaporation of $q$ = 2 m·y$^{-1}$ ($\cong$6·10$^{-8}$ m·s$^{-1}$) (see Sec. 1) the value of



$\gamma \equiv \dfrac{q}{h}\dfrac{x}{u} \cong 0.1$. This value is small enough to justify a Taylor series expansion of the Right
Hand Side (RHS, hereafter) of Eq. (2) for $\gamma \ll 1$, namely $1/(1-\gamma) \cong 1+\gamma$. Rearranging the
terms that result from this expansion of the RHS of Eq. (2) yields the following simple
expression:
$$\frac{S(x)-S(0)}{S(0)} \cong \gamma \tag{3}$$
Eq. (3) describes the first order relationship between the relative changes in salinity and the
non-dimensional parameter $\gamma$.

For a linearly varying salinity $S(x)=a \cdot x+b$, (where $a$ is the slope, $b$ is the intercept and

$x$ is the distance from the origin) the value of $\gamma$ is estimated from Eq. (3) as:
$$\gamma = \frac{S(x)-S(0)}{S(0)} = \frac{(ax+b)-b}{b} = \frac{ax}{b} \tag{4}$$

2.2. Changes in isotopic composition
The oxygen isotopic composition of water undergoing evaporation during the travel within a
semi enclosed basin can be described using the Rayleigh equation (e.g., Craig et al., 1956;
Appelo and Postma, 2005) as follows:
$$\delta^{18}O_{water}(x) = \delta^{18}O_{water}(0) + \varepsilon_{v,w} \ln(f) \tag{5}$$
where $\delta^{18}O_{water}$ $(x)$ and $\delta^{18}O_{water}$ $(x)$ are the oxygen isotopic composition of the seawater [‰]
at distance x and at the origin respectively, expressed in $\delta$ notation, such that
$\delta^{18}O_{samle} = \dfrac{R_{sample}-R_{s \tan derd}}{R_{sample}} \cdot 1000$, where $R_{sample}$ and $R_{standard}$ are the isotopic ratios (rare to
abundant isotope) of the sample and the standard respectively; $f$ is the fraction of liquid water
remaining after evaporation; $\varepsilon_{v,w} = 1000 \cdot (\alpha_{v,w}-1)$ is the enrichment factor [‰] and



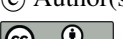

$\alpha_{v,w} = \dfrac{R_{vapor}}{R_{water}}$ is the isotope fractionation factor between vapor and water, where $R_{vapor}$ and
$R_{water}$ are the isotopic ratios of the vapor and water, respectively.
Using the Rayleigh equation, Eq. (5), a relationship between the hydrogen and oxygen
isotopic composition of water and the thermohaline circulation can be derived. Values for α
were computed using the data of Majoube (1971) (see also Rohling, 2007) with a temperature
of 28°C, which yields $\alpha_{\delta^{18}O} = 0.9909$ or $\varepsilon_{v,w} = -9.1$ ‰. Using the definition of $f$:

$$f = \frac{H_2{}^{16}O(x)}{H_2{}^{16}O(0)} = \frac{C_{H_2{}^{16}O(0)}V(x)}{C_{H_2{}^{16}O(0)}V(0)} = \frac{V(0)-Qt}{V(0)} = 1 - \frac{q}{h}\frac{x}{u} = 1 - \gamma \qquad (6)$$

where $H_2{}^{16}O(x)$ and $H_2{}^{16}O(0)$ are the weight of light water molecules at $x$ and at 0
respectively, $C_{H_2{}^{16}O(0)}$ is the concentration of light water molecules (g·cm$^{-3}$) and $V$ is the
volume of water [cm$^3$]. Substituting Eq. (6) into Eq. (5) yields:

$$\frac{\delta^{18}O_{water}(x) - \delta^{18}O_{water}(0)}{\varepsilon_{v,w}} = \ln(1-\gamma) \qquad (7)$$

For $(1-\gamma) \cong 1$, a Taylor expansion of $\ln(1-\gamma)$ near 1 yields the following relationship
between the non-dimensional parameter $\gamma$ and the isotopic composition of oxygen:

$$\frac{\delta^{18}O_{water}(x) - \delta^{18}O_{water}(0)}{\varepsilon_{v,w}} = -\gamma \qquad (8)$$

Similarly, one can derive the same relation for hydrogen isotopes by replacing in Eq.
(8) the variable $\delta^{18}O$ by $\delta D$ and the corresponding enrichment factor for oxygen isotopes by
that of hydrogen isotopes, $\varepsilon_{v,w} = -70.9$ ‰ as calculated from the fractionation factor for
hydrogen isotopes, $\alpha_{\delta D} = 0.9291$ (calculated from Majoube,1971, for temperature of 28°C).
For a linearly varying $\delta^{18}O$ (and similarly for $\delta D$) $\delta^{18}O(x)=a \cdot x+b$, (where $a$ is the
slope, $b$ is the intercept and $x$ is the distance from the origin) the value of $\gamma$ is estimated from
Eq. (8) as:



$$\gamma = \frac{\delta^{18}O(0) - \delta^{18}O(x)}{\varepsilon} = \frac{b - (ax + b)}{\varepsilon} = \frac{-ax}{\varepsilon} \qquad (9)$$

## 3. Application to the Red Sea and South-West Indian Ocean
The salt conservation equation, Eq. (1) and the isotopic fractionation equation, Eq. (5) have
both yielded simple expressions that involve the same nondimensional parameter $\gamma$. In order
to apply these relationships to particular basins the advection time $t_{adv}$ has to be large enough
to ensure that the relative changes in salinity and isotopic compositions are significant. In
order for this to hold, the ratio $x/u$ has to be large i.e. the water moves slowly and/or the
propagation distance is large. Depending on the length of the trajectory and the speed of the
flowing water, $t_{adv}$ can vary by orders of magnitude. In contrasts, the value of $q$ varies by a
factor of 2 only in regions far from the tropics (where precipitation is high) and the poorly
heated Poles. Thus, $q$ is of order 1 m/year ($\cong 3.15 \cdot 10^{-8}$ m·s$^{-1}$) in the subtropics and mid-
latitudes (see https://aquarius.umaine.edu/cgi/gal_images.htm?id=33) and since the depth of
the mixed layer $h$ of order of tens of meters, $t_{evp}=h/q$ varies by a factor of 2-3 only. We have
identified two basins where the required data is freely accessible and where these constraints
on large $t_{adv}$ and $t_{evp}$ that exceeds 1 year are satisfied. These regions are analyzed next.

### 3.1 Red Sea
#### 3.1.1. Hydrographic and Climatological Background
The circulation in the semi-enclosed Red Sea is strongly controlled by the net evaporation-
precipitation which is balanced by a net inflow through the straits of Bab el Mandeb that
connects the basin with the adjacent Indian Ocean. Though the currents associated with the
thermohaline circulation in semi-enclosed basins are fairly weak they provide the main
transport mechanism into the basin for heat, salt, nutrients and plankton. General reviews of
this important circulation with low-speed currents in semi-enclosed basins are given in



Pickard and Emery (1990) and Spall (2003) and specifically for the Red Sea in Zhai et al.

(2015).

Under steady state conditions, the amount of net evaporation can be estimated from

the difference in the salinity between the inflowing and outflowing waters by the Knudsen
relation. The outflowing water from the high salinity Red Sea to the Indian Ocean maintains
the steady state in salinity and water mass/volume inside the Red Sea. The time scale for
surface water to propagate from the straits of Bab el Mandeb to the northern tip of the Gulf of
Elat (Aqaba) is about 6 years (for $x$=2000 km and $u$=1 cm/s the travel time is $\cong 1.9 \cdot 10^8$ s ).
3.1.2. Data
The salinity and isotopic data used for the Red Sea were measured on seawater sampled
between October 23 and November 6, 1998 during cruises of R/V Sea Surveyor from the city
of Elat to the Seychelles archipelago (Fig. 1.a.; Steiner et al., 2014). Surface seawater samples
were collected underway every ~100 km using a specially designed water sampler (detailed
information on sampling, salinity and isotopic analyses and data acquisition in Steiner et al.,

2014).



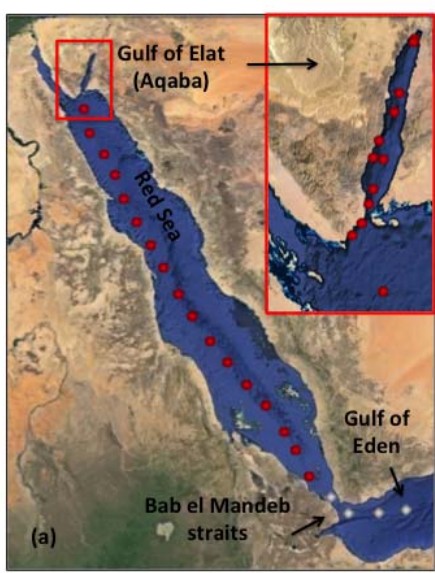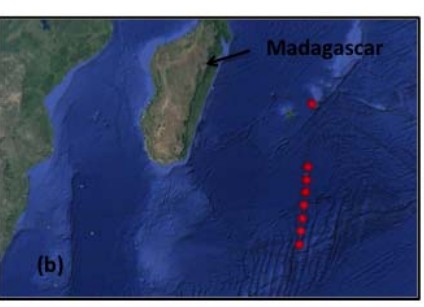


Figure 1: (a, left panel): Satellite image of the Red Sea with inset representing a zoom into the
Gulf of Elat (Aqaba) (adopted from Steiner et al., 2014). Red points represent Red Sea
sampling stations and white points represent Bab el Mandeb and Gulf of Aden stations (see
text for explanation). Salinity, $\delta^{18}O$ and $\delta D$ values for surface waters of Red Sea-Gulf of
Aden were taken from Steiner et al., 2014. (b, right panel): Satellite image of the Southern
Indian Ocean showing surface waters sampling stations (red dots). Salinity and $\delta^{18}O$ values
were taken from Srivastava et al., 2007.
High Correlation between $\delta^{18}O$, $\delta D$ and salinity has been previously reported in the Red Sea
by Craig (1966) and Steiner et al. (2014). The value of the three variables, salinity, $\delta^{18}O$ and
$\delta D$ were each plotted against their distance from Bab el Mandeb (x) and linear regression
lines were calculated to obtain intercepts and slopes for calculating the non-dimensional
parameter $\gamma$ (Fig. 2, Eq. (4) and Eq. (9), Table 1). The value of each variable at $x=0$ ($y$
intercept) was constrained from the average of 4 measurements close to the Bab el Mandeb
straits (white points in Fig. 1). Since the time scale of this flow is 6 years ($\cong 1.9 \cdot 10^8$ s)
(Section 3.1.1.), the measurements during the 2-week cruise in which the entire 2,000 km of
the length of the sea was sampled represent annually averaged values. Moreover, the
similarity in the $\delta^{18}O$ and $\delta D$ versus salinity plots for the years 1962 (Craig, 1966) and 1998



(Steiner et al., 2014), a 36 years period, suggests that the cross-section of these parameters
along the Red Sea represents a long-term steady state.
3.1.3. Results
Figure 2 shows the linear best fit of changes in salinity, $\delta^{18}O$ and $\delta D$ with distance from Bab-
el-Mandab. These best fitting lines have $R^2= 0.89$ for salinity, $R^2 = 0.85$ for $\delta^{18}O$ and $R^2=0.76$
for $\delta D$. The value of $\gamma$ is calculated from the intercepts and slopes of the best fitting lines from
Eq. (4) and Eq. (9) and the resulting estimates for $\gamma$ are given in Table 1. The confidence
intervals of these estimates were calculated from the uncertainty in the regression slopes. The
values of the non-dimensional parameter, $\gamma$, calculated from the oxygen and hydrogen
isotopic data agree very well with each other and both of these estimates fall within the 95%
confidence interval of salinity (Table 1, third column) with an average value of 0.1.

Sensitivity of the calculated values to the enrichment factor ($\alpha$) of $\delta^{18}O$ was tested by

using two different methods: 1) testing sensitivity to the specific formula by which the
enrichment factor is calculated using Dansgaard (1964; see section 1.2) instead of Majoube
(1971); and 2) changing the mean temperature used to calculate the enrichment factor in both
formulas (Majoube, 1971; Dansgaard, 1964) in the range of $28\pm1.5^{o}C$, the annual mean
temperature of the Red Sea (Nandkeolyar et al., 2013). Both methods show that the values
found for the non-dimensional parameter $\gamma$ are robust and the uncertainty due to the
variations in temperature or formulae used for the calculations results in variations that are
smaller than the confidence interval given in Table 1.



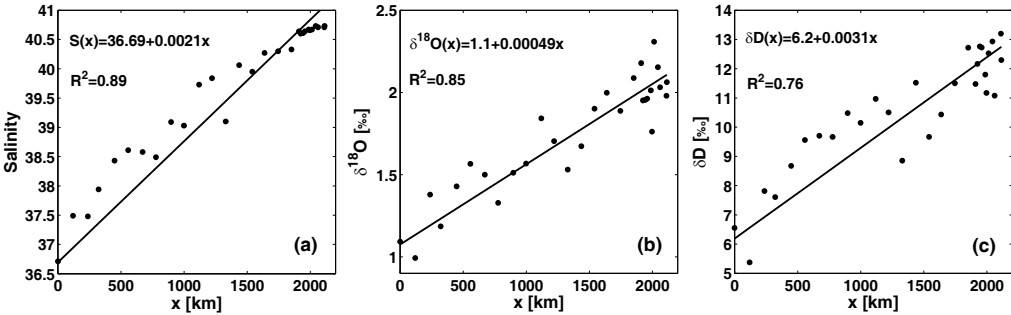


Figure 2: Salinity (a), $\delta^{18}O$ (b) and $\delta D$ (c) as a function of the distance from the straits of Bab
el Mandeb ($x$).
Table 1: Estimates of the non-dimensional parameter $\gamma$, based on salinity, $\delta^{18}O$ and $\delta D$. The
values of this parameter in column 3 were calculated from the slopes and intercepts of the
corresponding best-fit equations: Eq. (4) and Eq. (9) for salinity and $\delta^{18}O$, respectively and an
equation similar to Eq. (9) but for $\delta D$ instead of $\delta^{18}O$. x is the distance from the Bab el
Mandeb straits.

| Proxy variable | Best fit from data in Fig. 2 | $\gamma$ and 95% confidence interval |
|---|---|---|
| Sea Surface Salinity | $S(x)=36.69+0.0021x$ | $0.11\pm0.01$ |
| $\delta^{18}O$ | $\delta^{18}O(x)=1.1+0.00049x$ | $0.12\pm0.02$ |
| $\delta D$ | $\delta D(x)=6.2+0.0031x$ | $0.09\pm0.02$ |


3.2 The South-West Indian Ocean
3.2.1. Hydrographic and Climatological background
The subtropical anti-cyclonic Indian Ocean gyre consists of two western boundary currents,
the strong Agulhas current west of Madagascar and a weaker East Madagascar current. We
have located a second suitable study area, where the necessary data is available, east of the
East Madagascar Current where the water flows southward. This 20°S-30°S segment at ~57 °E
is located on the border between the tropics and subtropics (Figure 1). In contrast to the semi
enclosed Red Sea, where data has been collected for many years, the selected segment in the
South-West Indian Ocean is a small part of the subtropical gyre, and the details of the
hydrographic conditions along it are not known with sufficient accuracy (Stramma and



Lutjeharms, 1997). As a result, the annual velocities, mixed layer depth and evaporation rate
are less certain in this segment compared with those of the Red Sea. Since this is not a
western boundary current we assume that the mean velocities here are sufficiently small (no
more than a few centimeters per second) so that the advection time scale along this 1000 km
long segment is more than 1 year. Since along this segment evaporation exceeds precipitation
and the advection time is of the order of at least 1 year we treat this segment as suitable for
our analyses. The variation in isotopic composition and salinity along this segment is rather
small because the area north to latitude 20°S is relatively close to the Tropics and hence
excess evaporation is rather small (see e.g.
https://aquarius.umaine.edu/cgi/gal_images.htm?id=33) and the direction of flow is not
consistently southward.
3.2.2. Data
Salinity and $\delta^{18}O$ data reported in Srivastava et al., 2007 were downloaded from the World
Ocean Atlas (https://www.nodc.noaa.gov) for the region between 20°S-30°S and at ~57°E
(Fig. 1b). Similar to the Red Sea, high correlation between $\delta^{18}O$ was reported along this
segment by Srivastava et al. (2007), who suggested that the changes in isotopic ratio are
related with the excess of evaporation over precipitation along this segment. These specific
latitudes were chosen since they lie south of the Tropics (where precipitation exceeds
evaporation) and north of the North Sub-Tropical Front (south of which lie the Sub-Tropical
Mode Water) (see details in figure 1 of Srivastava et al., 2007).
3.2.3. Results
Figure 3 shows the linear best fit of changes in $\delta^{18}O$ and salinity along the segment: the
quality of the best fitting lines is given $R^2$= 0.91 for $\delta^{18}O$ and $R^2$ = 0.87 for salinity. Table 2
summarizes the corresponding values of $\gamma$ in the two calculations and the confidence intervals



of the estimates from the two variables barely overlap giving an average $\gamma$ value of 0.02 about
5 fold smaller than the $\gamma$ of the Red Sea.

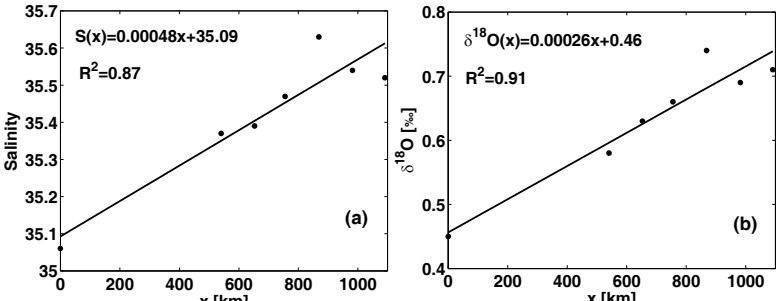

Figure 3: Salinity (a) and $\delta^{18}O$ (b) as a function of the distance from the most northern point
in 5$^o$N.

Table 2: Estimates of the non-dimensional parameter $\gamma$ based on salinity and $\delta^{18}O$ in the Indian
Ocean. The values of this parameter in column 3 were calculated from the slopes and
intercepts of the corresponding best-fit equations: Eq. (4) and Eq. (9) for salinity and $\delta^{18}O$. x
is the distance southward of 20$^o$S.

| Proxy variable | Best fit from data in Fig. 2 | $\gamma$ and 95% confidence interval |
|---|---|---|
| Sea Surface Salinity | $S(x)=35.09+0.00048x$ | 0.01±0.01 |
| $\delta^{18}O$ | $\delta^{18}O(x)=0.46+0.00026x$ | 0.03±0.01 |


4. Discussion

In this paper we derive, for the first time, a non-dimensional parameter $\gamma$ that estimates the
"strength" of the thermohaline circulation. The parameter $\gamma$ arises naturally from the equations
that express the conservation of such quantities as salt or certain isotopes that change in
response to intense evaporation. The non-dimensional parameter $\gamma$ combines four dimensional
parameters that govern the thermohaline circulation: the velocity of the flow ($u$); the depth of
the mixed layer ($h$); the length of water trajectory ($x$); and the evaporation rate along the
trajectory ($q$). Thus, when $\gamma$ is known any 3 of these dimensional parameters provides an




estimate of the fourth. It was shown in section 2.1 that $\gamma$ is the $t_{adv}/t_{evp}$ ratio, which are the time
to pass a typical distance by advection ($t_{adv}$) and the time to completely evaporate the mixed
layer ($t_{evp}$).

By the use of data collected from two different areas (from hydrological viewpoint) – the

Red Sea and the South-West Indian Ocean – we find that the resulting values of $\gamma$ from three
independent variables of: salinity, $\delta^{18}O$ and $\delta D$ are nearly identical. By the use of typical
scales for the Red Sea (for $x$=2000 km, $h$=100 m and $u$=1 cm·s$^{-1}$), our estimation of the excess
evaporation, $q$, are 1.8±0.2 m·y$^{-1}$ for salinity variations, 1.7±0.3 m·y$^{-1}$ from $\delta^{18}O$ variations
and 1.4±0.2 m·y$^{-1}$ from $\delta D$ variations. Despite the simplicity of this model and neglecting
other important physical processes, these estimates are in excellent agreement with previously
calculated estimates in the same region of 1.6-1.8 m·y$^{-1}$ (e.g., Tragou, 1999; Ben-Sasson et al.,
1999). Comparing the estimates that result from the various variables leads to the conclusion
that the simpler indirect methods yield consistent estimates that differ only slightly from the
elaborate direct method of detailed heat budget balance. Similarly, although the parameters in
the Indian are less certain than in the Red Sea, for $x$=1000 km, $h$=100 m and $q$=1 m·y$^{-1}$ the
estimate for the advection velocity is of order 1-3 cm·s$^{-1}$ i.e. an advection time of order 1 year.

The isotopic data complements the salinity data and can be used either for validating

estimates based on salinity changes or to substitute them. Though the accuracy of *in-situ*
salinity measurement is higher than that of isotopic ratios our results (Figs. 2, 3) suggest that
both variables yield similarly correlated cross-sections that are equally appropriate for
deriving reliable estimates of $\gamma$ in diverse oceanographic settings. However, from a practical
viewpoint and given present day technology it is much easier to measure salinity than isotopic
ratios. When one recalls the tremendous improvement that took place in the last decades in
the ease (and accuracy) of measuring salinity it is not unreasonable to expect that in the
coming decades similar improvement will take place in the ease (and accuracy) of measuring



isotopic ratios. At that time it will be possible to use our method routinely for verifying
salinity-based estimates of net evaporation.
To appreciate the physical meaning of the non-dimensional parameter $\gamma$ we envision the
following two extreme scenarios, or asymptotes of $\gamma$. First consider the case of very small $\gamma$,
i.e. when advection is very strong such as the western boundary currents where salinity
differences along the trajectory are too small (i.e. the water propagates too fast for the salinity
signal to be meaningful). In this case, $t_{adv}$ is in the order of days, (for $u=1$ m·s$^{-1}$ and $x=1000$
km), while $t_{evp}$ is of order 100 years in the most extreme case ($q=1$ m·y$^{-1}$ and $h=100$m) so the
corresponding value of $\gamma$ is ~$10^{-5}$. In this scenario, water flow is not controlled by excess
evaporation. The second scenario is that of $\gamma=1$ where the evaporation time equals the
advection time, i.e. when all the water in the mixed layer is evaporated while flowing over the
typical distance $x$. In this scenario, the evaporation and advection along the trajectory are
highly correlated and the (thermohaline) flow is driven by the excess evaporation.
As in other non-dimensional parameters, $\gamma$ has a physical meaning that can be appreciated
by examining the ratio between the advection and evaporation times. The approach we
advocate is similar to that employed in deciphering the physical meaning of the Rossby
number, that quantifies the deviation of the flow from geostrophic balance (the smaller the
Rossby number is, the closer the flow is to geostrophic balance). The parameter $\gamma \ll 1$ is
viewed as a measure of the degree to which the flow is thermohaline: When $\gamma$ is tiny (~$10^{-5}$)
the flow is far from thermohaline as in Western Boundary currents while when $\gamma \leq 1$ the flow
is close to being purely thermohaline as in the Red Sea and the Indian Ocean. The difference
of about 5-fold between the values of $\gamma$ in the two ocean regions (see Tables 1 and 2) suggest
that the flow in Red Sea is closer to being thermohaline more than the flow in Indian Ocean,
i.e., that the thermohaline flow in the former is driven by evaporation more than that in the
latter. However, this interpretation is not conclusive and may be result from a combination of





the slightly different values of $x$ and $h$ in the two sites in addition to differences in the values
of $u$, $q$. If all four physical parameters in $\gamma<1$ are known (e.g. when a numerical model
provides their typical values) in a certain basin its value can be easily calculated to provide a
measure of the degree to which the circulation there is thermohaline. The usefulness of $\gamma$ will
be determined by the degree to which the scientific community finds it helpful in interpreting
oceanic observations.
ACKNOWLEDGMENTS
The authors wish to thank the director and staff of the Inter University Institute for Marine
Sciences in Eilat for organizing the cruise of RV Sea Surveyor during which which the Red
Sea data were collected. We acknowledge the Israel Science Foundation for continued
support that enabled obtaining the Red Sea data for this research.

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
