# Peer review of "1. Introduction"

_Ocean Science, 2017_

## Referee Comment (RC1) · Anonymous Referee #1 · 1 Dec 2017

In this work, the authors derive a new parameter (gamma) to quantify what they call the "strength" of the thermohaline circulation that would be valid for all regions where the flow in the surface layer is weak and driven by evaporation (L#90-91). According to L#117-118, gamma would be related to changes in salinity of the inflow water due to evaporation (??), if I understood correctly. This "strength" concept is not well defined in the paper and I have a lot of doubts about it. I suggest the authors work harder to give a clear physical definition for what they call "strength". There are bits and pieces spread throughout the MS.

Despite the fact that the subject could be of interest for highly evaporative, semi-

enclosed basins such as the Red Sea, most assumptions made to derive gamma are certainly not valid for open-ocean basins such as the western Indian Ocean. The science behind the MS, and its presentation are not what I would expect to see in a high-quality journal. The MS has several conceptual problems. For example, the concept of thermohaline circulation is not accurate. In the first line of the introduction, the authors state that evaporation minus precipitation drives the thermohaline circulations in general. While this statement may be somewhat acceptable in relation to the Red Sea, it is certainly wrong for the Atlantic Ocean, for example. I strongly suggest the authors read Wunch's paper in Science (Wunsch, C., 2002: What is the thermohaline circulation? Science, 298, 1179-1181).

My overall impression is that the authors were thinking about the Red Sea (i.e. A semi-enclosed basin), and generalized their concepts as being valid everywhere, including the open-ocean—but such sloppy language induced statements that are simply wrong.

The authors also show a complete unfamiliarity with the hydrology, circulation and modern literature related to the southwestern Indian Ocean. This region is dominated by the westward-flowing South Equatorial Current (17S-18S), the eastward-flowing South Indian Countercurrent (23S-26S), characterized by strong eddy activity. There is also an anticyclonic cell centered east of Madagascar as shown by altimetric and hydrographic data. Besides the fact that the assumptions made by the authors are not valid in the southwestern Indian Ocean (There are a lot of good papers about this region published in the last ten years), the authors use only eight measurements to compute statistical relationships, which is misleading at least.

I also noticed the lack of important recent references about evaporation in the Red Sea. For example, Bower and Farrar (2015) show two-years of in situ evaporation measurements taken in the northern Red Sea (Bower and Farrar (2015), Air–Sea Interaction and Horizontal Circulation in the Red Sea. In: N.M.A. Rasul and I.C.F. Stewart (Eds.), The Red Sea,  Springer Earth System Sciences, DOI 10.1007/978-3-662-45201-

1_19).

Broadly speaking, the MS is rich in problems, lack new findings, and no innovations. Additionally, the title is quite obscure. I do not want to overwhelm the authors with such negative comments. My suggestion is to focus on the Red Sea, where some of the assumptions may be valid, and delete the western Indian Ocean part that compromised the MS with a large amount of errors. Instead of that, why not look at other semi-enclosed basins, such as the Mediterranean Sea? I believe the authors can find data there, maybe with contributions from other researchers. I would re-write completely the abstract and the introduction. It is a scientific manuscript, and therefore accurate statements are vital.

---

## Referee Comment (RC2) · Anonymous Referee #2 · 19 Dec 2017

The concept of 'strength of the thermohaline circulation' and the simple model behind can look appealing for those who are interested in simple explanations for complex problems – which is the essence of modelling –. However, the real meaning of this parameter $\gamma$ is not clear. Contrary to the claim of the title, this parameter cannot help quantifying thermohaline circulations.

First of all, thermohaline circulation is related to the water circulation induced by buoyancy gradients and is driven by heat and freshwater fluxes, while the authors consider evaporation only. The concept behind $\gamma$ could be meaningful in those regions that are dominated by evaporation but is certainly not applicable to the general thermohaline

circulation.

The concept of 'strength of the thermohaline circulation' could be related to the ratio of evaporation vs horizontal transport but has nothing to do with the physical mechanisms driving the thermohaline circulation.

Second, there are conflicting hypothesis in the discussion shown by the authors. On the one hand, they consider only those regions where the water moves slowly. But, on the other hand, they completely ignore diffusion against advection. This is not consistent. Evaporation should be compared with more robust transport time scales.

Third, the conclusions rely on a very few experimental data and lack therefore any statistical significance.

I would suggest that the authors drop the first part of their title (*Quantifying thermohaline circulations*) and focus on the second part (*seawater isotopic compositions and salinity as proxies of the ratio between transport time scale and evaporation time*) using a more extensive data sets from a single region.

---

## Author Comment (AC1) · 15 Jan 2018

General Both reviewers raised a number of issues that are totally acceptable to us and which we intend to ‎address in the revised version of the manuscript. These comments entail re-focusing the manuscript ‎on semi-enclosed basins and eliminating the use of the term thermo-haline circulation (which will ‎be changed in the revised manuscript to evaporation-driven circulation) since this term is presently ‎used in the context of the global circulation in the Atlantic. As part of the change in focus, we ‎intend to include in the revised version an analysis of surface salinity snapshots in the ‎Mediterranean that bolster our use of the proposed new non-dimensional

parameter. We thank the ‎reviewers for their insightful review that helped us better clarify the points we are trying to make.

Our response to each of the particular points raised by the reviewers is listed below.‎

Reviewer 1‎ ‎1. The term "strength" will not be used in the revised version. Instead $\gamma$ will be called the index of ‎evaporation driven circulation (EDC index). The magnitude of $\gamma$ (<1) determines the fraction of the ‎mixed layer (mean depth h) that evaporated during the time when the water moved a distance x at a ‎speed u. Thus, the larger the $\gamma$, the larger the contribution of the evaporation-driven circulation to ‎the actual circulation. The highest EDC index (measured in the extremely arid semi-enclosed Red ‎Sea) is 0.1. This will be explained in the revised version.‎ ‎2. Indeed the assumptions employed in the derivation of $\gamma$ are satisfied in semi-enclosed basins (the ‎Red Sea and the Mediterranean, RSM) and not in the open ocean. However, in the central part of ‎the Western branch of the Indian Ocean Gyre the numbers work very well (which is not at all the ‎case in other Western Boundary currents). We plan to focus the manuscript's sermon on RSM ‎and then to mention (separate from the "Results" section) that the idea works well in the Indian ‎Ocean even though not all assumptions are met.‎ ‎3. Additional references on the rates of evaporation in the Red Sea will be added ‎4. The title will be changed to: "Circulation in semi-enclosed basins: Quantifying the fraction of the ‎mixed layer that evaporates during the horizontal flow" so as to emphasize that the manuscript focuses on semi-enclosed basins.‎ ‎5. Data on East-West changes in SSS in the Mediterranean will be incorporated in the revised ‎version. We will show that in the Mediterranean Sea the available data shows seasonal changes in ‎the value of $\gamma$ that reflect seasonal changes in excess evaporation (see Fig. 1 below).

Please also note the supplement to this comment:
https://www.ocean-sci-discuss.net/os-2017-58/os-2017-58-AC1-supplement.pdf

[Figure]

**Figure 1:** Meridionally averaged sea surface salinity in Mediterranean Sea as a function of distance from the Straits of Gibraltar ($x$) every 3 months during the year 2017. Data is taken from the SMAP (Soil Moisture Active Passive) Sea Surface Salinity (SSS) level 3 8-day running average[1]. The slopes of the trend lines are the highest during August and November (when precipitation is minimal, i.e. excess evaporation is maximal) and lowest in February and May when precipitation is largest (i.e. excess evaporation is lowest). The resulting values of $\gamma$ vary between 0.06-0.07 in August and November, and are ~0.04 in February and May (only ~60 % of summer $\gamma$). These values are consistent with our claim that high $\gamma$ values reflect higher contribution of evaporation driven circulation. It is also consistent with the highest value of $\gamma =$ of 0.09-0.12 we estimated for the Red Sea, which is located in an extremely arid desert region with virtually no freshwater input and hence evaporation driven circulation is high there.

1 Meissner, T. and F. J. Wentz, 2016: Remote Sensing Systems SMAP Ocean Surface Salinities [Level 2C, Level 3 Running 8-day, Level 3 Monthly], Version 2.0 validated release. Remote Sensing Systems, Santa Rosa, CA, USA. Available online at www.remss.com/missions/smap, doi: 10.5067/SMP20-3SPCS.

**Fig. 1.**

[Figure]

---

## Author Comment (AC2) · 15 Jan 2018

General Both reviewers raised a number of issues that are totally acceptable to us and which we intend to ‎address in the revised version of the manuscript. These comments entail re-focusing the manuscript ‎on semi-enclosed basins and eliminating the use of the term thermo-haline circulation (which will ‎be changed in the revised manuscript to evaporation-driven circulation) since this term is presently ‎used in the context of the global circulation in the Atlantic. As part of the change in focus, we ‎intend to include in the revised version an analysis of surface salinity snapshots in the ‎Mediterranean that bolster our use of the proposed new non-dimensional

parameter. We thank the ‎reviewers for their insightful review that helped us better clarify the points we are trying to make. ‎

Our response to each of the particular points raised by the reviewers is listed below.‎

Reviewer 2‎ ‎1. The term "strength" will not be used in the revised version. Instead $\gamma$ will be called the index of ‎evaporation driven circulation (EDC index). The magnitude of $\gamma$ (<1) determines the fraction of the ‎mixed layer (mean depth h) that evaporated during the time when the water moved a distance x at a ‎speed u. Thus, the larger the $\gamma$, the larger the contribution of the evaporation-driven circulation to ‎the actual circulation. The highest EDC index (measured in the extremely arid semi-enclosed Red ‎Sea) is 0.1. This will be explained in the revised version.‎ ‎2. While the flow we have in mind is driven by excess evaporation, changes in SSS are inherent to ‎the circulation (in fact, the change in surface salinity is the main indicator of our proposed theory). ‎The term "thermohaline" will be replaced by the term evaporation-driven circulation in the revised ‎version.‎ ‎3. Diffusion is indeed ignored in our theory which focuses on Evaporation-driven flows. In the ‎context of our gross estimates, molecular diffusion is entirely negligible on the O(1000) km length ‎scale, while no data can be invoked to estimate the eddy diffusion in these flows. A mean value of ‎mixed layer depth (h) is used throughout even though mixing processes change it as time goes by. ‎ ‎4. Additional data from the Mediterranean (including seasonal changes) will be added in the revised ‎version (see Fig. 1 below).‎ ‎5. The title will be modified to: "Circulation in semi-enclosed basins: Quantifying the fraction of ‎the mixed layer that evaporates during the horizontal flow"‎

Please also note the supplement to this comment:
https://www.ocean-sci-discuss.net/os-2017-58/os-2017-58-AC2-supplement.pdf

[Figure]

**Figure 1:** Meridionally averaged sea surface salinity in Mediterranean Sea as a function of distance from the Straits of Gibraltar ($x$) every 3 months during the year 2017. Data is taken from the SMAP (Soil Moisture Active Passive) Sea Surface Salinity (SSS) level 3 8-day running average[1]. The slopes of the trend lines are the highest during August and November (when precipitation is minimal, i.e. excess evaporation is maximal) and lowest in February and May when precipitation is largest (i.e. excess evaporation is lowest). The resulting values of $\gamma$ vary between 0.06-0.07 in August and November, and are ~0.04 in February and May (only ~60 % of summer $\gamma$). These values are consistent with our claim that high $\gamma$ values reflect higher contribution of evaporation driven circulation. It is also consistent with the highest value of $\gamma=$ of 0.09-0.12 we estimated for the Red Sea, which is located in an extremely arid desert region with virtually no freshwater input and hence evaporation driven circulation is high there.

1 Meissner, T. and F. J. Wentz, 2016: Remote Sensing Systems SMAP Ocean Surface Salinities [Level 2C, Level 3 Running 8-day, Level 3 Monthly], Version 2.0 validated release. Remote Sensing Systems, Santa Rosa, CA, USA. Available online at www.remss.com/missions/smap, doi: 10.5067/SMP20-3SPCS.

**Fig. 1.**